# Prognostic Role of miR-221 and miR-222 Expression in Cancer Patients: A Systematic Review and Meta-Analysis

**DOI:** 10.3390/cancers11070970

**Published:** 2019-07-11

**Authors:** Gloria Ravegnini, Sarah Cargnin, Giulia Sammarini, Federica Zanotti, Justo Lorenzo Bermejo, Patrizia Hrelia, Salvatore Terrazzino, Sabrina Angelini

**Affiliations:** 1Department of Pharmacy and Biotechnology, University of Bologna, 40126 Bologna, Italy; 2Department of Pharmaceutical Sciences and Interdepartmental Research Center of Pharmacogenetics and Pharmacogenomics (CRIFF), University of Piemonte Orientale, 28100 Novara, Italy; 3Institute of Medical Biometry and Informatics, University of Heidelberg, 69120 Heidelberg, Germany

**Keywords:** miR-221, miR-222, overall survival, progression-free survival, cancer

## Abstract

*Background:* A wealth of evidence has shown that microRNAs (miRNAs) can modulate specific genes, increasing our knowledge on the fine-tuning regulation of protein expression. miR-221 and miR-222 have been frequently identified as deregulated across different cancer types; however, their prognostic significance in cancer remains controversial. In view of these considerations, we performed an updated systematic review and meta-analysis of published data investigating the effects of miR-221/222 on overall survival (OS) and other secondary outcomes among cancer patients. A systematic search of PubMed, Web of Knowledge, and Cochrane Library databases was performed. Hazard ratios (HRs) and 95% confidence intervals (95% CIs) were used to assess the strength of association. *Results:* Fifty studies, analyzing 6086 patients, were included in the systematic review. Twenty-five studies for miR-221 and 17 studies for miR-222 which assessed OS were included in the meta-analysis. High expression of miR-221 and miR-222 significantly predicted poor OS (HR: 1.48, 95% CI: 1.14–1.93, *p* = 0.003 and HR: 1.90, 95% CI: 1.43–2.54, *p* < 0.001, respectively). Subgroup analysis revealed that the finding on miR-221 was not as robust as the one on miR-222. Furthermore, high miR-222 expression was also associated with worse progression-free survival and disease-free survival pooled with recurrence-free survival. *Conclusions:* The meta-analysis demonstrated that high expression of miR-222 is associated with poor prognosis in cancer patients, whereas the significance of miR-221 remains unclear. More work is required to fully elucidate the role of miR-221 and miR-222 in cancer prognosis, particularly in view of the limitations of existing results, including the significant heterogeneity and limited number of studies for some cancers.

## 1. Introduction

MicroRNAs (miRNAs) are a wide family of small non-coding RNAs that have emerged as key players in regulating the expression of target genes and are therefore important in many biological processes [1]. In the last two decades, a wealth of evidence has clarified the mechanisms through which miRNAs can modulate specific genes, increasing our knowledge on the fine-tuning regulation of protein expression [1,2,3,4]. miRNAs have been shown to be involved in complex regulatory networks, basically playing a role in every aspect of the biology of cells and organs [5,6]. Therefore, it is clear that miRNA deregulation may promote the disruption of a fine-tuned equilibrium within a cell, altering physiologically relevant functions [7,8,9]. Consequently, it is not surprising that miRNA expression has been found to be dysregulated in the majority of human diseases, including cancer [10,11]. miRNA deregulation takes place via different mechanisms such as (i) deletion or amplification of miRNA genes [12,13]; (ii) aberrant expression of transcription factors [14]; (iii) dysregulation of the miRNA biogenesis process [15]; and (iv) miRNA sequestration (miRNA sponging) [16,17,18]. miRNA deregulation can affect the hallmarks of cancer, including sustaining proliferation, evading apoptosis and resisting cell death, promoting invasion and metastasis, and inducing angiogenesis [19]. In particular, miRNAs may act as oncogenes, tumor suppressors, or, depending on the cellular context, may have both functions [20]. In addition to the cancerogenetic process, compelling evidence has shown that miRNAs can be involved in tumor sensitivity to treatment, showing a correlation with patients’ clinical outcomes [21,22,23,24].

miR-221 and miR-222 have been frequently identified as deregulated across different cancer types [25,26,27,28,29,30]. These two miRNAs, encoded on chromosome X (Xp11.3), are highly homologous, share seed sequences, and usually act as a cluster (miR-221/222) [31,32]. In particular, this cluster might promote the overcoming of the status of cell quiescence and increase proliferation, survival, and metastatic potential, acting as an oncogene [31,33,34]. In addition, it is well documented that miR-221 and miR-222 play a key role in modulating the clinical outcome in cancer patients in both solid and hematological malignancies [30,35]. In this regard, the prognostic value of miR-221 and miR-222 expression in cancer has been extensively investigated in the last decade, with controversial results. A number of studies support the hypothesis that their overexpression may be predictive for poor cancer survival [36,37], while others reported a better prognosis [38,39] or failed to find a significant association [40,41]. Few meta-analyses have investigated the association between miR-221/222 [42,43,44] and malignancies; however, their prognostic role in cancer patients has not been fully analyzed. In addition, single studies may be underpowered to achieve a comprehensive and reliable conclusion. Given these premises, it is imperative and timely to perform an exhaustive meta-analysis to evaluate the prognostic value of miR-221/222 expression in cancer patients. Therefore, we systematically performed a meta-analysis based on all eligible studies to evaluate the association between miR-221/222 expression and the prognosis of patients with malignant tumors.

## 2. Methods

### 2.1. Search Strategy

The present systematic review was conducted in accordance with the Preferred Reporting Items for Systematic Reviews and Meta-analyses (PRISMA) Statement principles [45]. The PubMed, Web of Knowledge, and Cochrane Library databases were systematically searched for original articles analyzing the prognostic value of miR-221/222 in different cancer patients (last updated search 28 September 2018). Relevant studies were selected using Boolean combinations of the following key terms variably combined: “miR-221 OR miR-222 OR miRNA-221 OR miRNA-222” OR miR-221/222 AND “cancer OR tumor OR tumour OR neoplasia OR tumors OR tumours OR cancers” AND “survival OR prognosis OR progression”. Additionally, the reference list of review papers, meta-analyses, and all original studies were hand searched to acquire further relevant studies missed by the initial electronic quest.

### 2.2. Inclusion and Exclusion Criteria

Eligible studies were required to meet the following inclusion criteria: (i) miR-221/222 expression measured in the tumor tissue; (ii) investigating the correlation of miR-221 or miR-222 expression level with human cancer prognosis; and (iii) reporting hazard ratios (HR) with corresponding 95% confidence intervals (CI) or sufficient information to estimate them. Exclusion criteria were (i) circulating miRNA, case reports, review articles, and editorials; (ii) non-human and in vitro studies; and (iii) non-English articles. When more than one study had been published sharing part of the same patient population, only the most recent and complete study was selected in this meta-analysis.

### 2.3. Data Extraction

After removing duplicated studies, two independent investigators (G.S. and F.Z.) carefully read the titles and abstracts of the relevant articles and judged their eligibility. Then, the entire text of potential eligible studies was evaluated to assess appropriateness for inclusion in the meta-analysis. Disagreements were resolved by discussion with another investigator (S.T. or S.A.) for consensus. Two authors (G.R. and S.C.) independently extracted the data from all included studies. The following data were sought and recorded: (1) first author, publication year, and nationality of studied population; (2) miRNA analyzed, cancer type, sample type, and miRNA assay method; and (3) sample size, cut-off value, HR, and corresponding 95% CI. When a study reported the survival results of both univariate and multivariate analyses, only the latter was extracted, as it accounts for confounding factors and is therefore more accurate. For studies only providing Kaplan–Meier curves, variables were read from the graphical survival plots using PlotDigitizer software (version 2.6.8, freely available at http://plotdigitizer.sourceforge.net); after that the HR value and 95% CI were estimated via the method described by Parmar et al. [46]. Finally, extracted data were crosschecked by two investigators (G.R. and S.C.), and any divergences were resolved by discussion with a third author (S.T. or S.A.).

### 2.4. Outcomes and Definition

In the present study, all survival outcomes were extracted. Overall survival (OS) was the primary outcome of interest selected to calculate the association with miR-221 and/or miR-222 expression in cancer patients. OS was defined as the length of time from when the sample was obtained to the date of death from any cause or the date of last follow-up. All the others were considered secondary survival outcomes.

### 2.5. Quality Assessment

The scientific quality of the included studies was evaluated according to the Newcastle–Ottawa Scale (NOS) for nonrandomized studies [47]. The methodological quality of included studies was firstly evaluated by two investigators (G.S. and F.Z.) and then by two other investigators (G.R and S.C.) in a second round of study quality evaluation; any disagreement was resolved through consensus. The NOS, assigning up to a maximum of 9 points, consists of three components: (I) selection and definition of the study groups (0–4 points); (II) comparability of the cohorts (0–2 points); and (III) ascertainment of outcomes (0–3 points). A possible score of 0–9 was assigned to each study. Studies with a NOS score greater than 7 were considered of high quality.

### 2.6. Statistical Analysis

Analyses were performed using ProMeta software Version 2 (INTERNOVI di Scarpellini Daniele s.a.s., Cesena FC, Italy). The pooled HR and corresponding 95% CI were used to evaluate the prognostic value of miR-221 and/or miR-222 for different malignancies. All analyses were performed using the low miR-221/222 expression group as the reference group (HR = 1). An observed HR of >1 implies a worse prognosis for the group with higher miR-221/222 expression. Heterogeneity between studies was estimated using the χ^2^-based Cochran’s Q statistic (significant for *p* < 0.10) [48], and the *I*^2^ index (range 0%–100%) which quantifies heterogeneity irrespective of the number of studies (an *I*^2^ of over 50% is interpreted as the presence of large degree of heterogeneity) [49]. Survival outcomes were pooled using the random effects model which takes into account both within-study variance and between-study variance; the random effects model coincides with the fixed effect model in the absence of heterogeneity (*I*^2^ = 0) [34]. To assess the robustness of the overall findings, subgroup analyses were conducted based on multiple criteria, including patient ethnicity, tumor type, biological specimen, sample size, survival analysis, and NOS. Additionally, the presence of small-study effects and publication bias was evaluated graphically by funnel plots and statistically by means of Egger’s test [50]. An Egger’s test *p* value of <0.10 was considered statistically significant. Moreover, leave-one-out meta-analysis was performed to identify studies that had a crucial influence on the pooled HR by removing one study at a time.

## 3. Results

### 3.1. Study Characteristics and Quality Assessment

The study selection process on the PubMed, Web of Knowledge, and Cochrane databases yielded 852 studies. After removing 242 duplicated publications, the remaining 610 records were evaluated by carefully reading titles and abstracts, after which 248 studies were excluded for the following reasons: only in vitro studies, reviews or meta-analyses, non-human studies, not in English, and case reports. Then, the entire text of the remaining 362 studies was assessed, determining the removal of 312 further studies. Finally, 50 articles, published between 2009 and 2018, were included in the systematic review [35,36,37,38,39,40,41,51,52,53,54,55,56,57,58,59,60,61,62,63,64,65,66,67,68,69,70,71,72,73,74,75,76,77,78,79,80,81,82,83,84,85,86,87,88,89,90,91,92,93]. A detailed flow-chart illustrating the literature selection process is presented in Figure 1.

The main general information on the included studies are reported in Table 1 and Table 2. Among them, 27 focused on miR-221 [36,38,40,52,54,57,60,61,62,64,69,70,72,73,74,75,79,80,81,82,83,84,85,88,89,91,92], 11 on miR-222 [37,39,53,56,59,63,67,68,86,87,93], and the remaining 12 investigated the cluster miR-221/miR-222 [35,41,51,55,58,65,66,71,76,77,78,90]. Overall, 6086 patients suffering from different tumors were included in the systematic review, with the sample size in each study ranging from 20 to 393 patients. With regard to the cancer type, three studies focused on hematological tumors [56,62,92], whereas all the others evaluated solid tumors. Overall, the most evaluated neoplasms in the present meta-analysis were hepatocellular carcinoma (HCC; 7 studies) [54,64,71,80,83,91,93], non-small cell lung cancer (NSCLC; 5 studies) [37,67,79,84,86], prostate cancer (PCa; 5 studies) [51,52,60,66,76], breast cancer (BC; 4 studies) [61,75,81,87], and colorectal cancer (CRC; 4 studies) [38,40,72,90]. The majority of the studies were performed in Asia (32/50, 64%; 25 in China) and Europe (10/50, 20%). All the studies were retrospective in design; qRT-PCR was mostly used to measure the expression of miR-221/222 (45/50 studies, 90%). Twenty-four out of 50 studies (48%) measured the expression in formalin-fixed, paraffin-embedded (FFPE) samples, 21 (42%) in fresh/frozen tissue, 2 studies used both specimens, another 2 studies (4%) did not specifically report the tissue type, and 1 used bone marrow. The cut-off value stratifying patients into high and low expression groups varied among the different studies, with the median value being the most broadly used value (26/50 studies, 52%). The length of follow-up ranged from 24 to 300 months. Regarding the quality of included studies, the NOS scores ranged from 5 to 9 (median 7). Thirty-six studies that had scores of ≥7 were categorized as high-quality studies [35,36,37,38,39,40,41,51,52,54,56,57,59,60,61,62,65,66,67,68,69,70,71,72,73,75,77,78,79,80,84,85,87,89,90,92], while the remaining 14, which had scores of <7, were categorized as low-quality studies [53,55,58,63,64,74,76,81,82,83,86,88,91,93].

### 3.2. Impact of miR-221 Expression on Survival Outcomes

With regard to miR-221, 25 [35,36,38,40,41,55,57,58,62,64,69,71,72,73,78,79,80,82,83,84,85,88,89,90,92] out of the 50 studies included in the systematic review evaluated the relationship between miR-221 expression and OS in patients suffering from different tumors. Three studies reported cancer-specific survival (CSS) [61,70,91] and one reported the disease-specific survival (DSS) [77]. Concerning secondary outcomes, six studies reported disease-free survival (DFS) [35,38,62,74,80,81]; five reported recurrence-free survival (RFS) [51,52,60,66,75]; four reported progression-free survival (PFS) [35,71,76,77]; one reported the distant metastasis rate (DMR) and local recurrence rate (LRR) [40]; and three studies reported the time to local recurrence (TTLR) [54], time to recurrence (TTR) [91], and metastasis-free survival (MFS) [65], respectively.

#### 3.2.1. miR-221 Expression and OS

We observed a high degree of heterogeneity among the 25 studies reporting OS and included in the meta-analysis (*I*^2^ = 75%, *p* < 0.001). Despite this, the pooled HR indicated a significantly short OS in patients with higher expression of miR-221 (N_patients_ = 2240; HR: 1.48, 95% CI: 1.14–1.93, *p* = 0.003; Figure 2A). No evidence of publication bias or small-study effects was observed in either the funnel plot (Figure 2B) or in Egger’s test (*p* = 0.770).

In order to assess the robustness of the overall finding, subgroup analyses were further performed based on seven subcategories: ethnicity, tumor type, pre-operative treatment, specimen, sample size, survival analysis (univariate or multivariate), and NOS score (Table 3). Statistically significant heterogeneity was detected in all the subgroup analyses except for studies performed in patients with liver cancer (*I*^2^ = 0%, *p* = 0.698) or NSCLC (*I*^2^ = 0%, *p* = 0.602) or with no pre-operative anticancer treatment (*I*^2^ = 0%, *p* = 0.813). When stratified by ethnicity, miR-221 overexpression was significantly correlated with poor OS in Asians with the combined HR being 1.67 (95% CI: 1.30–2.14, *p* < 0.001). When stratified by cancer types, high miR-221 expression was associated with poor OS in patients with liver cancer (pooled HR: 1.98, 95% CI: 1.51–2.60, *p* < 0.001) and NSCLC (pooled HR: 1.92, 95% CI: 1.31–2.81, *p* = 0.001). When stratified by pre-operative anticancer treatment, high miR-221 expression was associated with poor OS in patients with no pre-treatment (pooled HR: 1.84, 95% CI: 1.56–2.17, *p* < 0.001). In the subgroup analysis stratified by type of biological specimen, high miR-221 expression showed a significant relationship with poor OS in both FFPE samples (pooled HR: 1.68, 95% CI: 1.20–2.36, *p* = 0.003) and fresh/frozen tissue (pooled HR: 1.54, 95% CI: 1.05–2.28, *p* = 0.028). For the subgroup analysis of sample size, high expression of miR-221 in samples of <100 patients was significantly associated with poor OS (pooled HR: 1.62, 95% CI: 1.21–2.18, *p* < 0.001), whereas no association was observed in samples of equal to or greater than 100 patients. The analysis also showed that high expression of miR-221 was significantly associated with poor OS in both the survival methods. Finally, in subgroup analysis stratified by quality assessment, high miR-221 expression was associated with poor OS in high-quality studies, with the combined HR being 1.61 (95% CI 1.20–2.16, *p* = 0.002).

The four studies reporting CSS [61,70,91] and DSS [77] available for the meta-analysis were characterized by significant heterogeneity (*I*^2^ = 88%, *p* < 0.001). No significant association was found between CSS/DSS and high expression of miR-221, with the pooled HR being 0.95 (95% CI: 0.52–1.76, *p* = 0.878; Table 4). The forest plot is shown in Appendix A. We also analyzed OS along with CSS and DSS. Despite significant heterogeneity (*I*^2^ = 77%; *p* < 0.001), high miR-221 expression was significantly associated with poor OS/CSS/DSS (pooled HR: 1.38, 95% CI: 1.09–1.74, *p* = 0.007; Appendix A).

#### 3.2.2. miR-221 Expression and Secondary Outcomes

DFS was analyzed along with RFS by pooling together 11 studies [35,38,51,52,60,62,66,74,75,80,81] encompassing 913 cancer patients. We observed a large degree of heterogeneity (*I*^2^ = 84%, *p* < 0.001) and no significant association between disease progression and high expression of miR-221 (HR: 0.90, 95% CI: 0.46–1.75, *p* = 0.753; Appendix A). No evidence of publication bias or small-study effects was observed in either the funnel plot (Appendix A) or in Egger’s test (*p* = 0.901). When assessing the association of miR-221 with PFS, no significant results were found (Table 4).

### 3.3. Impact of miR-222 on Survival Outcomes

Among the 23 studies focusing on the prognostic value of miR-222 in cancer patients, 17 studies investigated the effect of miR-222 overexpression on OS [35,37,39,41,53,55,58,59,63,65,67,68,71,78,86,90,93]. Only one study reported DSS [77] in place of OS. Concerning secondary outcomes, seven studies were available for the meta-analysis of miR-222 and PFS [35,55,56,71,76,77,86], three studies investigated its correlation with DFS [35,87,93], two with RFS [51,66], and one study reported MFS [65].

#### 3.3.1. miR-222 Expression and OS

We observed a high degree of heterogeneity among the 17 studies reporting OS and included in the meta-analysis (*I*^2^ = 77%, *p* < 0.001). Despite this, the pooled results of miR-222 and OS provided evidence of significantly poorer prognosis in patients expressing higher levels of miR-222 (*N*_patients_ = 1782; HR: 1.90, 95% CI: 1.43–2.54, *p* < 0.001; Figure 3A). There was no evidence of publication bias either from the funnel plot (Figure 3B) or from Egger’s test (*p* = 0.299). Subgroup analyses were further performed to evaluate the robustness of the overall finding on the association between miR-222 expression and OS, based on the same seven categories used for miR-221; the data are presented in Table 5. Statistically significant heterogeneity was detected in most of the subgroups except for studies performed in patients with liver cancer (*I*^2^ = 0%, *p* = 0.441), gastro-intestinal malignancies (*I*^2^ = 25%, *p* = 0.256), or NSCLC (*I*^2^ = 0%, *p* = 0.470) or studies with a NOS score of <7 (*I*^2^ = 0%, *p* = 0.602). The results confirmed the significant association between high miR-222 expression and poor OS in all the subgroup analyses conducted, with the exception of studies including patients with urogenital cancer (HR: 1.22, 95% CI: 0.29–5.07, *p* = 0.781) and those conducted on FFPE tissues (HR: 1.22, 95% CI: 0.79–1.88, *p* = 0.360).

We also analyzed OS along with DSS; despite significant heterogeneity (*I*^2^ = 75%, *p* < 0.001), high miR-221 expression was significantly associated with poor OS/DSS (pooled HR: 1.88, 95% CI: 1.46–2.42, *p* < 0.001, Appendix A).

#### 3.3.2. miR-222 Expression and Secondary Outcomes

Despite a large degree of heterogeneity (*I*^2^ = 73%, *p* = 0.001), the results showed a significantly shorter PFS in patients with high expression of miR-222 (HR: 1.67, 95% CI: 1.10–2.51, *p* < 0.015; Appendix A). No evidence of publication bias or small-study effects was observed in either the funnel plot (Appendix A) or in Egger’s test (*p* = 0.914). When assessing the association of miR-222 with DFS pooled with RFS, high expression was associated with poor prognosis (Table 4; Appendix A).

### 3.4. Sensitivity Analysis

For both miR-221 and 222, sensitivity analysis of the OS was performed to investigate the influence of each individual study on the pooled HRs. The analysis (Appendix A, respectively) showed that the pooled results were not significantly altered by omitting any single data set. This result suggests that no single study significantly influenced the pooled HRs or the 95% CIs, highlighting that the pooled results of OS were robust.

## 4. Discussion

In the last two decades, compelling evidence has repeatedly demonstrated the key involvement of miRNAs in cancer [19]. miRNAs minutely regulate a plethora of biological mechanisms at the post-transcriptional level, and an increasing number of studies have provided clues that aberrantly expressed miRNAs are involved in diagnosis and prognosis in numerous chronic diseases, including cancer [94,95]. Specifically, miRNAs may have both oncogenic and tumor-suppressive roles in any step of cancerogenesis, depending on the cellular context and tumor type. Among the miRNAs frequently dysregulated in cancer, miR-221 and miR-222 are considered of great importance. Indeed, the miR-221/222 cluster acts as *onco-miR* in the majority of epithelial tumors [96] while acting as *oncosuppressor-miR* in erythroleukemic cells [97]. Recently, diverse reports have shown their prognostic importance as circulating molecules in plasma, blood, and other biological fluids (for a review on circulating molecules, see ref [98]). Furthermore, one of the emerging relevant aspects is the contribution of miR-221 and miR-222 to the clinical outcome in cancer patients. To the best of our knowledge, to date, only two meta-analyses have assessed the association of miR-221 and miR-222 with the prognosis of patients with malignant tumors, respectively encompassing 1204 [44] and 2693 [42] patients. Results from both the meta-analyses agree on the association between miR-221/222 overexpression and poor OS and their possible translation into clinical practice. However, both meta-analyses were based on a limited number of studies. In particular, with regard to the primary outcome, few studies were published at the time of the first meta-analysis [40], which included nine studies for miR-221 and four studies for miR-222. In the meta-analysis performed by Zhang and coworkers [42], the authors pooled together miR-221 and miR-222; therefore, they suggested the development of biomarkers in cancer prognosis based on the cluster. However, some studies highlighted opposite results on survival outcomes between the two miRNAs; thus, a different prognostic role for miR-221 and miR-222 cannot be ruled out. In addition, Zhang et al. [42]., combined different sample sources (i.e., tissue and blood), adding a degree of uncertainty to the conclusive result. In view of this evidence, we performed an updated meta-analysis to corroborate the previous conclusions and possibly uncover some novel findings on secondary outcomes, separately analyzing the prognostic roles of these two miRNAs. We identified 50 studies, published between 2009 and 2018, including a total of 6086 patients, covering 21 different tumor types, and representing the widest meta-analysis so far performed. The current study demonstrated that high expression of both miR-221 and miR-222 significantly predicted poor OS in cancer, confirming the findings of the two previous meta-analyses [42,44]. Subgroup analysis revealed that the finding on miR-221 was not as robust as the one on miR-222. Indeed, high miR-222 expression was associated with poor OS, irrespectively of ethnicity, sample size, and quality score. On the contrary, miR-221 was a predictor of poor OS only in Asian (not in Caucasian) patients, in small cohorts of patients (<100), and in high-quality studies (NOS ≥ 7). With regard to ethnicity, differences in the genetic background, dietary habits, and environmental exposure might in part explain the findings. However, the fact that significance is retained only in studies of small sample size makes the result unconvincing, and we cannot exclude the possibility that the significance of miR-221 is due to chance. Additionally, subgroup analysis based on tissue type did not reveal differences in the prognostic value of miR-221. On the contrary, high miR-222 expression was significantly associated with poor OS in fresh/frozen tissue but not in FFPE samples. The finding is consistent with the knowledge of a higher speed of RNA degradation in FFPE tissues than in the other two specimen types [99]. With regard to secondary outcome, high expression of miR-222 was significantly associated with both worse PFS and worse DFS/RFS, whereas no association was observed with miR-221. Unfortunately, due to the limited number of studies investigating secondary outcomes, we could not perform a subgroup analysis to verify the robustness of the findings on miR-222. Despite this, the results on miR-222 are convincing, as they are consistent with OS. Overall, the findings have important clinical implications, as they may provide a rationale for the study and the development of a molecular sponge to regulate miR-222 expression [18,100]. In this context, intensive research is currently ongoing to understand the role of miRNA sponges that could be used to control miRNA expression and therefore function for therapeutic purposes [101,102].

Although the present meta-analysis provides evidence of the prognostic value of miR-222, several limitations should be highlighted. Primarily, although 50 relevant studies were identified, the number of studies for some cancer types was insufficient to be evaluated. Therefore, when analyzing OS, subgroup analysis was carried out by considering the anatomic site instead of the specific type of cancer. In this context, high miR-222 expression was associated with poor OS in all the anatomic sites of cancer with the exception of urogenital cancer. In view of this consideration, we should not underestimate the limitations associated with differences in patients’ baseline characteristics, treatments received, duration of follow-up, and cancer stage as reported also for other miRNAs [81,85,89,103,104,105]. Additional bias could arise from differences in the biological properties of the specific cancer types, particularly regarding the tumor microenvironment and the diversity of signals provided by the tumor cells. Further sources of micro statistical errors could be (i) the variety of cut-off values used to define miR-221/222 expression among studies; (ii) the calculation of HR and 95% CI from the survival curve in some studies; and (iii) the statistical methodology, i.e., the use of univariate in place of multivariate analysis in some investigations. Therefore, better-designed clinical studies are needed to gain new insights into the prognostic role of miR-222 in different human malignancies. In particular, to reduce the source of heterogeneity and make meta-analysis a powerful tool, we should also consider reaching an agreement on the clinical outcomes. Indeed, of the 23 studies investigating the prognostic role of miR-222, only 17 investigated OS, while the others explored the association with different secondary outcomes; these could not be analyzed together, reducing the number of studies eligible for the meta-analysis.

## 5. Conclusions

In conclusion, to the best of our knowledge, our study represents the widest meta-analysis so far performed on the prognostic value of miR-221 and miR-222. These findings should bring the attention of scientists to the importance of high miR-222 expression as a biomarker of poor prognosis in terms of both OS and secondary outcomes, including DFS and RFS. On the contrary, the results on miR-221 remain controversial. Despite the fact that no obvious publication bias was detected in the analysis, we cannot exclude that what we observed is a chance finding. Therefore, further large-scale prospective studies are warranted to validate the appropriateness of miR-222’s poor prognosis prediction capability and to exclude the involvement of miR-221.

## Figures and Tables

**Figure 1 cancers-11-00970-f001:**
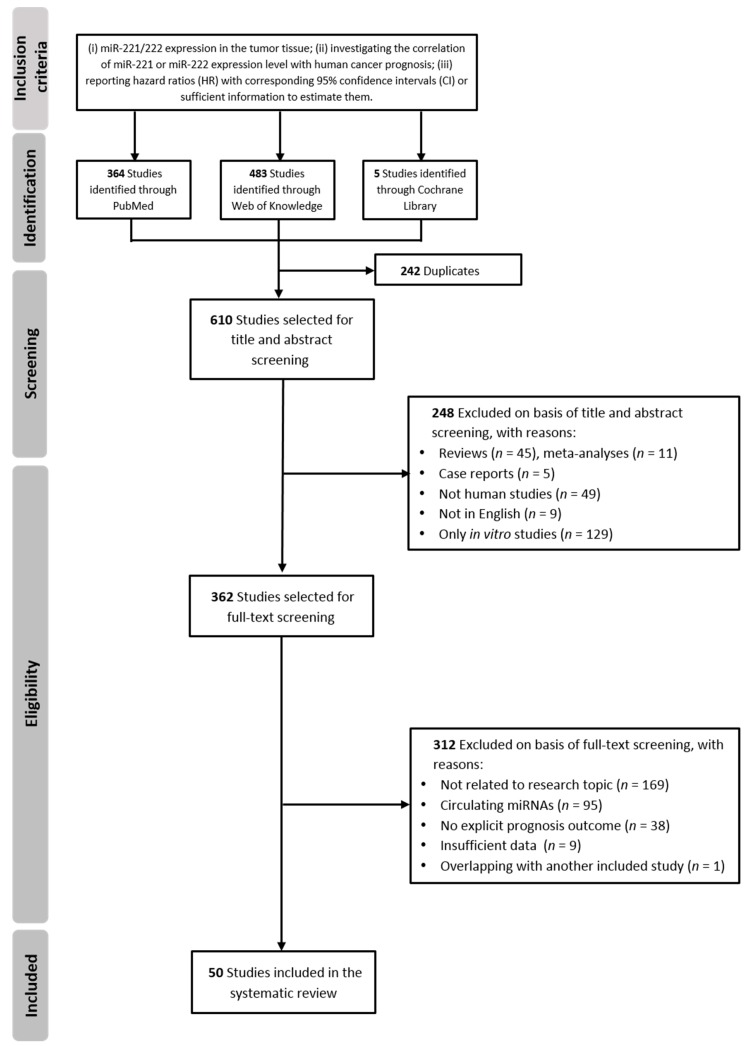
Flowchart of the literature search and selection process of eligible studies.

**Figure 2 cancers-11-00970-f002:**
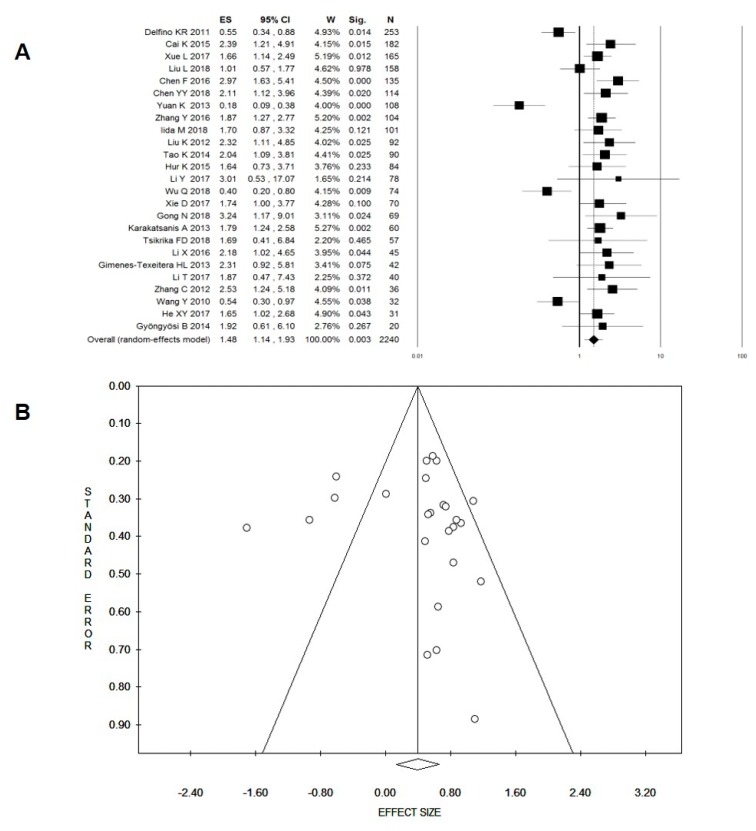
Forest (**A**) and Funnel (**B**) plots for the effect of miR-221 overexpression on overall survival. ES, effect size (i.e., hazard ratio); W, weight; Sig, statistical significance; N, total number of cancer patients included in the survival analysis.

**Figure 3 cancers-11-00970-f003:**
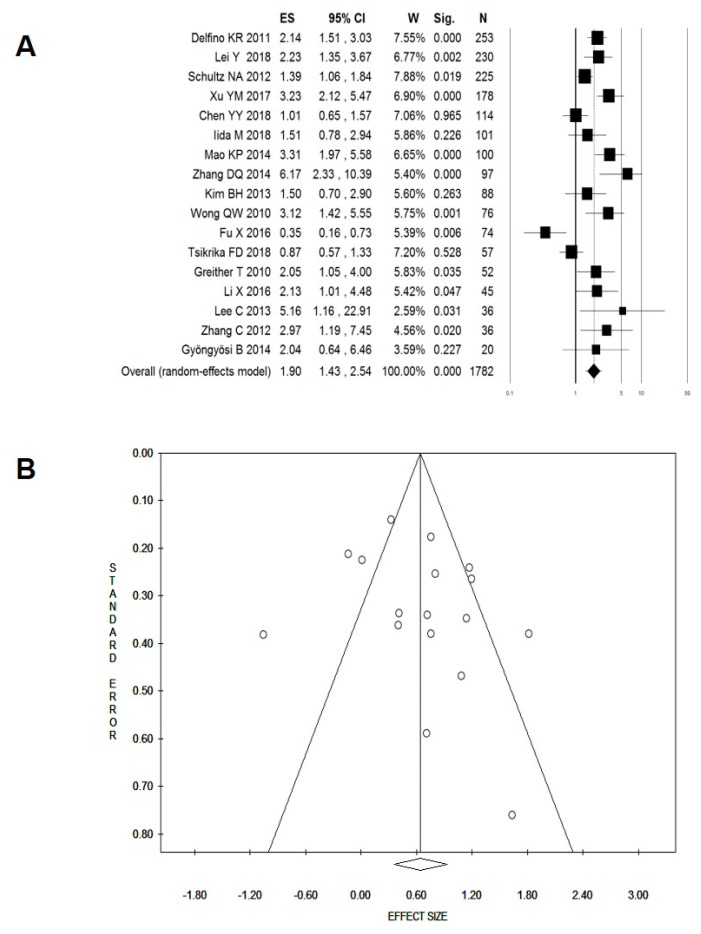
Forest (**A**) and Funnel (**B**) plots for the effect of miR-222 overexpression on overall survival. ES, effect size (i.e., hazard ratio); W, weight; Sig, statistical significance; N, total number of cancer patients included in the survival analysis.

**Table 1 cancers-11-00970-t001:** Main characteristics of studies evaluating the impact of higher expression of miR-221 or miR-222 on survival outcomes in cancer patients.

First Author [Ref] Year	Country	Sample Size (Pts Analyzed ^#^)	Gender (M/F)	Age Median (Range) or Mean ± SD or Age Category	Cancer Type	Clinical Classification	Pre-Operative (Localized or Systemic) Treatment	Type of Specimen	NOS Score
Gramantieri et al. [91] 2009	Italy	(51) 45	40/11	68 (49–82)	HCC	ES stage: G2–G4	No	frozen tissue/FFPE	5
Wang et al. [92] 2010	China	85 (32 ALL)	42/43	43 (14–82)	ALL/AML	FAB classification; ALL: L1–L3, Tcell; AML: M0–M6, mast cell	NR	bone marrow	7
Wong et al. [93] 2010	China	99 (76)	80/19	57 (48–66.5)	HCC	T stage: T1–T3	No (NAT in 3 pts)	frozen tissue	5
Schaefer et al. [51] 2010	Germany	76 (75)	76/0	63 (49–74)	PCa	Tumor stage: pT2–pT3	No	frozen tissue	8
Spahn et al. [52] 2010	Germany	92	92/0	67 (52–78)	PCa	Tumor stage: pT2–pT4	No	FFPE	9
Greither et al. [53] 2010	Germany; UK	56 (52)	34/22	61.7 (34–80)	PC	NR	NR	frozen tissue	6
Yoon et al. [54] 2011	Korea	115	91/24	51 (26–76)	HCC	TNM stage: I–III	No	FFPE	7
Delfino et al. [55] 2011	USA	253	162/91	55.7	GBM	WHO glioma grade 4	NS	NR	6
Alencar et al. [56] 2011	USA; Canada; Spain	176	NR	59.5 (16–92)	Lymphoma	AAS: I–IV	No	FFPE	9
Liu et al. [57] 2012	China	92	50/42	59.3 ± 5.6	GC	TNM stage: I–III	No	frozen tissue	8
Zhang et al. [58] 2012	China	72 (36)	NR	NR	GM	Grade: I–IV	NR	FFPE/frozen tissue	6
Schultz et al. [59] 2012	Denmark	328 (225)	111/114	64 (31–85)	PC	WHO stage: I–II	NR	FFPE	8
Kang et al. [60] 2012	Korea	92 (73)	92/0	64.7 (50–76)	PCa	TNM stage: T2–T3	NS	FFPE	8
Hanna et al. [61] 2012	USA	473	NR	NR	BC	NR	NR	FFPE	8
Gimenes-Teixeira et al. [62] 2013	Brazil	48 (42)	39/9	20.1 (1–66)	T-ALL	NR	NR	Fresh/frozen tissue	7
Lee et al. [63] 2013	China	60 (36)	38/22	≤60 y (*n* = 18); >60 y (*n* = 42)	PC	Stage: I–IV	NR	frozen tissue	6
Yuan et al. [38] 2013	USA	108	68/40	<65 y (*n* = 63) ≥65 y (*n* = 45)	CRC	T stage: T2–T4	NR	frozen tissue	7
Karakatsanis et al. [64] 2013	Greece	81 (60 HCC)	60/21	≤60 y (*n* = 21) >60 y (*n* = 60)	HCC/ICC	T stage: T1–T4	No	FFPE	6
Kim et al. [65] 2013	Korea	91 (MFS: 79; OS: 88)	57/34	61 (25–80)	GC	Tumor stage: pT2–pT4	No	FFPE	7
Amankwah et al. [66] 2013	USA	65 (miR-221: 63; miR-222: 60)	65/0	59.5 (46–75)	PCa	Stage: I–IV	NR	FFPE	7
Mao et al. [67] 2014	China	100	59/41	<60 y (*n* = 34) ≥60 y (*n* = 66)	NSCLC	TNM stage: I–III	NR	frozen tissue	8
Zhang et al. [68] 2014	China	97	51/46	<60 y (*n* = 59); ≥60 y (*n* = 38)	Bladder	T stage: Ta–T1 (*n* = 82); ≥T2 (*n* = 15)	No	fresh/frozen tissue	8
Tao et al. [69] 2014	China	90	47/43	<70 y (*n* = 42); ≥70 y (*n* = 46); unknown (*n* = 2)	CC	TNM stage: I–IV	No	FFPE	7
Vergho et al. [70] 2014	Germany	74 (37)	48/26	66.8	RCC	T stage: T1–T3; Tumor grade: G1–G4	NR	FFPE	8
Gyöngyösi et al. [71] 2014	Hungary	20	16/4	68 (52–82)	HCC	Advanced stage	No	FFPE	7
Hur et al. [72] 2015	USA	393 (84)	66/18	64.6 ± 10.7 (*n* = 84)	CRC	TNM stage: I–IV	NR	frozen tissue	8
Cai et al. [73] 2015	China	182	73/109	52 ± 6.25	CC	TNM stage: I–III	No	FFPE	8
Li et al. [74] 2015	China	25	14/11	<60 y (*n* = 13); ≥60 y (*n* = 12)	EHCC	Stage: I–IV	No	frozen tissue	5
Eissa et al. [75] 2015	Egypt	78 (76)	0/78	52.28 ± 12	BC	Stage: I-III; grade: I–III	No	frozen tissue	9
Goto et al. [76] 2015	Japan	54 (52)	54/0	73.5 (58–88)	PCa	TNM stage: III–IV	NR	NR	6
García-Donas et al. [77] 2016	Belgium	145 (74)	90/48	61.0 (35–82)	RCC	Metastatic	immuno- or chemotherapy	FFPE	8
Li et al. [78] 2016	China	45	30/15	<50 y (*n* = 18); ≥50 y (*n* = 27)	GM	TNM stage: I–IV; T classification: T1–T4	NR	fresh tissue	8
Zhang et al. [79] 2016	China	104	58/46	<55 y (*n* = 44); ≥55 y (*n* = 60)	NSCLC	TNM stage: I–IV	No	frozen tissue	7
Fu et al. [39] 2016	China	74	0/74	<50 y (*n* = 37); ≥50 y (*n* = 37)	OC	FIGO stage: I–IV	NR	FFPE	8
Chen et al. [80] 2017	China	135	76/59	53.3 ± 9.1	HCC	Tumor stage: I–II	No	FFPE	7
Deng et al. [81] 2017	China	125	0/125	51.6 (25–81)	BC	T stage: T1–T4	No	FFPE	6
He et al. [82] 2017	China	31	16/15	2.9 (0.1–12)	NeuroB	Tumor stage IIb–IV	No	FFPE	5
Xue et al. [36] 2017	China	165	114/51	<60 y (*n* = 132) ≥60 y (*n* = 33)	GM	Tumor stage: I–IV	NR	frozen tissue	7
Li et al. [83] 2017	China	40	18/22	<60 y (*n* = 17); >60 y (*n* = 23)	HCC	NR	No	frozen tissue	5
Li et al. [84] 2017	China	78	52/26	<63 y (*n* = 38); ≥63 y (*n* = 40)	NSCLC	TNM stage: I–IV	No	frozen tissue	8
Xie et al. [85] 2017	China	70	38/32	<50 y (*n* = 29); ≥50 y (*n* = 41)	Liver	Tumor stage: I–IV	No	frozen tissue	8
Xu et al. [86] 2017	China	178	93/85	<60 y: *N* = 90; ≥60 y: *N* = 88	NSCLC	Grade: Low: Ia–IIa; high: IIb–IIIa	No	frozen tissue	6
Han et al. [87] 2017	Korea	197	0/197	49 (27–85)	BC	T stage: T1–T3	NR	FFPE	8
Wu et al. [88] 2018	China	107 (74)	0/107	<50 y (*n* = 37); ≥50 y (*n* = 37); unknown (*n* = 74)	OC	FIGO stage: I–IV	NR	FFPE	6
Gong et al. [89] 2018	China	69	38/31	<40 y (*n* = 51); ≥40 y (*n* = 18)	OsteoS	Tumor stage: II–III	NR	FFPE	8
Lei et al. [37] 2018	China	230	113/117	<55 y (*n* = 114); ≥55 y (*n* = 116)	NSCLC	TNM stage: I–IV	No	FFPE	7
Liu et al. [40] 2018	China	158	95/63	56.9 ± 7.4	CRC	TNM stage: I–IV	No	FFPE	9
Tsikrika et al. [35] 2018	Greece	182 (DSF/PFS: 101; OS: 57)	152/30	70	Bladder	Tumor stage: pTa, pT1–pT4 WHO grade: 1–3	No	frozen/fresh tissue	9
Iida et al. [90] 2018	Japan	113 (101)	49/52; unknown (*n* = 12)	68.0 ± 11.6	CRC	Tumor stage: I–II e IV	No	frozen tissue	7
Chen et al. [41] 2018	Taiwan	114	68/46	54 (38–65)	GBM	WHO glioma grade 4	NR	FFPE	8

# Patients analyzed. Abbreviations: AAS, Ann Arbor System of staging; ALL, acute lymphoblastic leukemia; AML, acute myeloid leukemia; BC, breast cancer; CC, colon cancer; CRC, colorectal cancer; DFS, disease-free survival; EHCC, extrahepatic cholangiocarcinoma; ES; Edmondson and Steiner’s criteria; FFPE, formalin-fixed, paraffin-embedded tissue; FIGO: International Federation of Obstetricians and Gynecologists; GC, gastric cancer; GBM, glioblastoma; GM, glioma; HCC, hepatocellular carcinoma; ICC, intrahepatic cholangiocarcinoma; MFS, metastasis-free survival; NA, not applicable; NAT, neoadjuvant therapy; NeuroB: Neuroblastoma; NOS, Newcastle–Ottawa Scale; NR, not reported; NS, not specified; NSCLC, non-small cell lung cancer; OC, ovarian cancer; OS, overall survival; OsteoS: Osteosarcoma; PC, pancreatic cancer; PCa, prostate cancer; PFS, progression-free survival; RCC, renal cell carcinoma; SD, standard deviation; T-ALL, T-cell acute lymphoblastic leukemia; TNM, tumor nodes and metastases staging system.

**Table 2 cancers-11-00970-t002:** Characteristics and prognostic information extracted from the studies included in the systematic review.

First Author [Ref] Year	miRNA Analyzed and Clinical Outcome Extracted	Detection Method	Cut-Off Value	Clinical Outcome: HR (95% CI)
Gramantieri et al. [91] 2009	miR-221: TTR ^u^/CSS ^u^	qRT-PCR	median	TTR: 1.73 (1.15–2.60) *; CSS: 1.27 (0.64–2.52) *
Wang et al. [92] 2010	miR-221: OS ^m^	qRT-PCR	median	OS: 0.54 (0.30–0.97)
Wong et al. [93] 2010	miR-222 OS ^u^/DFS ^u^	qRT-PCR	fold change ≥ 2	OS: 3.12 (1.42–5.56); DFS: 2.21 (1.19–3.83)
Schaefer et al. [51] 2010	miR-221: RFS ^m^;miR-222: RFS ^m^	qRT-PCR	median	RFS: 0.50 (0.01–39.1); RFS: 5.04 (0.03–9.40)
Spahn et al. [52] 2010	miR-221: RFS ^m^	qRT-PCR	median	RFS: 0.53 (0.29–0.95)
Greither et al. [53] 2010	miR-222: OS ^m^	qRT-PCR	median	OS: 2.05 (1.05–4.00)
Yoon et al. [54] 2011	miR-221: TTLR ^m^	qRT-PCR	mean	TTLR: 3.07 (1.56–6.07)
Delfino et al. [55] 2011	miR-221: OS ^m^ miR-222: OS ^m^/PFS ^m^	Microarray	NR	OS: 0.55 (0.34–0.88); OS: 2.14 (1.51–3.03); PFS: 1.44 (1.11–1.86)
Alencar et al. [56] 2011	miR-222: PFS ^m^	qRT-PCR	median	PFS: 2.26 (1.24–4.14)
Liu et al. [57] 2012	miR-221: OS ^m^	qRT-PCR	mean	OS: 2.32 (1.11–4.85)
Zhang et al. [58] 2012	miR-221: OS ^u^ miR-222: OS ^u^	IHC	final score ≤ 3	OS: 2.53 (1.24–5.18) *; OS: 2.97 (1.19–7.74) *
Schultz et al. [59] 2012	miR-222: OS ^m^	microarray	10th percentile	OS: 1.39 (1.06–1.84)
Kang et al. [60] 2012	miR-221: RFS ^u^	qRT-PCR	median	RFS: 0.36 (0.17–1.90)
Hanna et al. [61] 2012	miR-221: CSS ^m^	qRT-PCR	75th percentile	CSS: 0.70 (0.51–0.97)
Gimenes-Teixeira et al. [62] 2013	miR-221: OS ^m^/DFS ^m^	qRT-PCR	median	OS: 2.31 (0.92–5.81); DFS: 1.54 (0.57–4.17)
Lee et al. [63] 2013	miR-222: OS ^m^	qRT-PCR	median	OS: 5.16 (1.16–22.91)
Yuan et al. [38] 2013	miR-221: OS ^m^/DFS ^m^	qRT-PCR	NR	OS: 0.18 (0.09–0.16); DFS: 0.04 (0.01–0.16)
Karakatsanis et al. [64] 2013	miR-221: OS ^m^	qRT-PCR	mean	OS: 1.79 (1.24–2.58)
Kim et al. [65] 2013	miR-221: MFS ^u^ miR-222 OS ^m^/MFS ^u^	qRT-PCR	fold change ≥ 3	MFS: 1.72 (1.03–2.89) *; OS: 1.50 (0.70–2.90); MFS: 1.92 (1.05–3.50) *
Amankwah et al. [66] 2013	miR-221: RFS ^m^ miR-222: RFS ^m^	qRT-PCR	median	RFS: 1.79 (0.67–4.76); RFS: 2.56 (0.87–7.14)
Mao et al. [67] 2014	miR-222: OS ^m^	qRT-PCR	median	OS: 3.31 (1.97–5.58)
Zhang et al. [68] 2014	miR-222: OS ^m^	qRT-PCR	median	OS: 6.17 (2.33–10.39)
Tao et al. [69] 2014	miR-221: OS ^m^	qRT-PCR	median	OS: 2.04 (1.09–3.81)
Vergho et al. [70] 2014	miR-221: CSS ^m^	qRT-PCR	1.84 of ROC curve	CSS: 0.47 (0.22–1.00)
Gyöngyösi et al. [71] 2014	miR-221: OS ^u^/PFS ^u^ miR-222: OS ^u^/PFS ^u^	qRT-PCR	median	OS: 1.92 (0.61–6.10); PFS: 1.32 (0.47–3.66); OS: 2.04 (0.64–6.46); PFS: 1.43 (0.51–3.99)
Hur et al. [72] 2015	miR-221: OS ^m^	Microarray	1.68 of ROC curve	OS: 1.64 (0.73–3.71)
Cai et al. [73] 2015	miR-221: OS ^m^	qRT-PCR	median	OS: 2.39 (1.21–4.91)
Li et al. [74] 2015	miR-221: DFS ^u^	qRT-PCR	median	DFS: 2.19 (1.07–4.47) *
Eissa et al. [75] 2015	miR-221: RFS ^m^	qRT-PCR	1.03 of ROC curve	RFS: 14.84 (1.89–116.25)
Goto et al. [76] 2015	miR-221: PFS ^u^ miR-222: PFS ^m^	qRT-PCR	NR	PFS: 0.69 (0.42–1.14) *; PFS: 0.21 (0.07–0.64)
García-Donas et al. [77] 2016	miR-221: DSS ^m^/PFS ^m^ miR-222: DSS ^m^/PFS ^m^	NGS and qRT-PCR	median	DSS: 1.71 (1.35–2.16); PFS: 2.25 (1.67–3.02); DSS: 1.77 (1.41–2.22); PFS: 2.02 (1.51–2.71)
Li et al. [78] 2016	miR-221: OS ^u^ miR-222: OS ^u^	qRT-PCR	mean	OS: 2.18 (1.02–4.65); OS: 2.13 (1.01–4.48)
Zhang et al. [79] 2016	miR-221: OS ^m^	qRT-PCR	mean	OS: 1.87 (1.27–2.77)
Fu et al. [39] 2016	miR-222: OS ^m^	qRT-PCR	median	OS: 0.35 (0.16–0.73)
Chen et al. [80] 2017	miR-221: OS ^m^/DFS ^m^	qRT-PCR	median	OS: 2.97 (1.63–5.41); DFS: 2.85 (1.56–5.18)
Deng et al. [81] 2017	miR-221: DFS ^m^	qRT-PCR	median	DFS: 0.48 (0.26–0.88)
He et al. [82] 2017	miR-221: OS ^u^	qRT-PCR	mean	OS: 1.65 (1.02–2.68)
Xue et al. [36] 2017	miR-221: OS ^u^	qRT-PCR	median	OS: 1.66 (1.13–2.49)
Li et al. [83] 2017	miR-221: OS ^u^	qRT-PCR	NR	OS: 1.87 (0.47–7.43) *
Li et al. [84] 2017	miR-221: OS ^m^	qRT-PCR	median	OS: 3.01 (0.53–17.07)
Xie et al. [85] 2017	miR-221: OS ^m^	qRT-PCR	NR	OS: 1.74 (1.00–3.77)
Xu et al. [86] 2017	miR-222: OS ^m^/PFS ^u^	qRT-PCR	mean	OS: 3.23 (2.12–5.47); PFS: 2.62 (1.56–4.39) *
Han et al. [87] 2017	miR-222: DFS ^m^	qRT-PCR	median	DFS: 5.67 (1.08–29.76)
Wu et al. [88] 2018	miR-221: OS ^u^	qRT-PCR	median	OS: 0.39 (0.20–0.80)
Gong et al. [89] 2018	miR-221: OS ^m^	qRT-PCR	median	OS: 3.24 (1.17–9.01)
Lei et al. [37] 2018	miR-222: OS ^m^	IHC	final score ≥ 5	OS: 2.23 (1.35–3.67)
Liu et al. [40] 2018	miR-221: OS ^m^/LRR ^m^/DMR ^m^	qRT-PCR	Youden index of ROC curve	OS: 1.01 (0.57–1.77); DMR: 1.21 (0.66–2.20); LRR: 1.17 (0.65–2.11)
Tsikrika et al. [35] 2018	miR-221: OS ^u^/DFS ^u^/PFS ^u^ miR-222: OS ^u^/DFS ^m^/PFS ^m^	qRT-PCR	30th percentile	OS: 1.69 (0.41–6.84) *; DFS: 0.71 (0.38–1.33); PFS: 1.37 (0.54–3.62); OS: 0.87 (0.57–1.33) *; DFS: 2.59 (1.15–5.83); PFS: 8.98 (1.10–73.54)
Iida et al. [90] 2018	miR-221: OS ^u^ miR-222: OS ^u^	qRT-PCR	median	OS: 1.70 (0.87–3.32); OS: 1.51 (0.78–2.94)
Chen et al. [41] 2018	miR-221: OS ^m^ miR-222: OS ^m^	qRT-PCR	fold change > 1	OS: 2.11 (1.12–3.97); OS: 1.01 (0.64–1.58)

^m^ Multivariate; ^u^ univariate; * data extracted from the Kaplan–Meier curve. Abbreviations: CSS, cancer-specific survival; DFS, disease-free survival; DMR, distant metastasis rate; DSS, disease-specific survival; HR, hazard ratio; IHC, immunohistochemistry; LRR, local recurrence rate; MFS, metastasis-free survival; NGS, next-generation sequencing; NR, not reported; OS, overall survival; PFS, progression-free survival; qRT-PCR, quantitative reverse transcription polymerase chain reaction; RFS, recurrence-free survival; ROC, receiver operating characteristic; TTLR, time to local recurrence; TTR, time to recurrence.

**Table 3 cancers-11-00970-t003:** Summary of random effect meta-analyses for the association between miR-221 overexpression and overall survival of cancer patients.

Study Groups	Studies Included	Test of Association		Test of Heterogeneity	
		HR (95% CI)	*p*-Value	*I*^2^%	*p*-Value
**All studies**	25	1.48 (1.14–1.93)	0.003	75	<0.001
**Ethnicity**					
Asian	18	1.67 (1.30–2.14)	<0.001	62	<0.001
Mostly Caucasian	5	1.30 (0.69–2.44)	0.410	75	0.003
Mixed	2	0.64 (0.05–7.72)	0.723	94	<0.001
**Type of cancer**					
Liver	5	1.98 (1.51–2.60)	<0.001	0	0.698
Gastro-intestinal	7	1.29 (0.68–2.43)	0.434	83	<0.001
Neurological	6	1.56 (0.98–2.48)	0.062	77	<0.001
Urogenital	2	0.72 (0.18–2.90)	0.638	70	0.069
NSCLC	2	1.92 (1.31–2.81)	0.001	0	0.602
Other	3	1.51 (0.45–5.01)	0.504	84	0.002
**No pre-treatment ^#^**	14	1.84 (1.56–2.17)	<0.001	0	0.813
**Sample detected**					
Fresh/frozen tissue	12	1.54 (1.05–2.28)	0.028	70	<0.001
FFPE	10	1.68 (1.20–2.36)	0.003	66	0.002
**Sample size ***					
≥100	9	1.27 (0.78–2.08)	0.332	86	<0.001
<100	16	1.62 (1.21–2.18)	0.001	58	0.002
**Survival analysis**					
Univariate	9	1.54 (1.07–2.21)	0.020	55	0.024
Multivariate	16	1.46 (1.02–2.09)	0.039	81	<0.001
**NOS score**					
≥7	19	1.61 (1.20–2.16)	0.002	71	<0.001
<7	6	1.17 (0.65–2.11)	0.604	84	<0.001

* Number of cancer patients; ^#^ Patients without pre-operative anticancer treatment; FFPE, formalin-fixed, paraffin-embedded tissue; HR, hazard ratio; NOS, Newcastle–Ottawa Scale; NSCLC, non-small cell lung cancer.

**Table 4 cancers-11-00970-t004:** Summary of meta-analyses for the association between high miR-221/222 expression and survival outcomes in cancer patients.

Survival Outcome	Studies Included	Test of Association		Test of Heterogeneity		Egger’s *p*-Value
		HR (95% CI)	*p*-Value	*p*-Value Q-Test	*I*^2^%	
**miR-221**						
CSS	3	0.75 (0.48–1.17)	0.205	0.148	48	0.913
CSS/DSS	4	0.95 (052–1.76)	0.878	<0.001	88	0.448
DFS	6	0.79 (0.31–1.98)	0.614	<0.001	89	0.338
RFS	5	1.09 (0.37–3.17)	0.881	0.007	71	0.443
PFS	4	1.31 (0.65–2.63)	0.452	0.001	82	0.504
**miR-222**						
PFS	7	1.67 (1.10–2.51)	0.015	0.001	73	0.914
DFS	3	2.50 (1.58–3.94)	<0.001	0.573	0	0.053
DFS/RFS	5	2.52 (1.66–3.82)	<0.001	0.573	0	0.143

Abbreviations: CSS, cancer-specific survival; DFS, disease-free survival; DSS, disease-specific survival; HR, hazard ratio; OS, overall survival; PFS, progression-free survival; RFS, recurrence-free survival.

**Table 5 cancers-11-00970-t005:** Summary of random effect meta-analyses for the association between miR-222 overexpression and OS in cancer patients.

Study Groups	Studies Included	Test of Association		Test of Heterogeneity	
		HR (95% CI)	*p*-Value	*I*^2^%	*p*-Value
**All studies**	17	1.90 (1.43–2.54)	<0.001	77	<0.001
**Ethnicity**					
Asian	12	2.11 (1.40–3.17)	<0.001	79	<0.001
Mostly Caucasian	5	1.52 (1.07–2.18)	0.021	66	0.019
**Type of cancer**					
Liver	2	2.80 (1.55–5.04)	0.001	0	0.441
Gastro-intestinal	5	1.52 (1.22–1.90)	<0.001	25	0.256
Neurological	4	1.80 (1.12–2.90)	0.015	66	0.033
Urogenital	3	1.22 (0.29–5.07)	0.781	93	<0.001
NSCLC	3	2.88 (2.16–3.83)	<0.001	0	0.470
**No pre-treatment ^#^**	8	2.16 (1.37–3.42)	0.001	77	<0.001
**Sample detected**					
Fresh/frozen tissue	9	2.48 (1.60–3.85)	<0.001	77	<0.001
FFPE	6	1.22 (0.79–1.88)	0.360	73	0.002
Sample size *					
≥100	7	1.93 (1.40–2.66)	<0.001	74	0.001
<100	5	1.92(1.13–3.28)	0.016	81	<0.001
**Survival analysis**					
Univariate	6	1.82 (1.12–2.97)	0.016	64	0.015
Multivariate	11	1.94 (1.34–2.80)	<0.001	81	<0.001
**NOS score**					
≥7	11	1.58 (1.09–2.30)	0.016	80	<0.001
<7	6	2.56 (2.03–3.22)	<0.001	0	0.602

* Number of cancer patients; ^#^ Patients without pre-operative anticancer treatment; **FFPE**, formalin-fixed, paraffin-embedded tissue; **HR**, hazard ratio; **NOS**, Newcastle–Ottawa Scale; **NSCLC**, non-small cell lung cancer.

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
