# Peer review of "Prognostic Role of miR-221 and miR-222 Expression in Cancer Patients: A Systematic Review and Meta-Analysis"

_cancers, 2019, doi:10.3390/cancers11070970_

Round 1
Reviewer 1 Report
The author answered my questions.
Reviewer 2 Report
The reviewers have addressed all my concerns so I recommend acceptance of the paper.
Reviewer 3 Report
The authors have satisfactorily responded to all my questions and made the necessary changes to the manuscriptThis manuscript is a resubmission of an earlier submission. The following is a list of the peer review reports and author responses from that submission.
Round 1
Reviewer 1 Report
The manuscript presented by Ravegnini et al provided a review and Meta analysis of miR-221 and miR-222 in cancer prognosis, which could be of interest to the clinic researchers. Although miR221/222 has been recognized as tumor marker, either tumor suppressor or oncogene, this review is quite different from the others. The Meta analysis could provide direct evidence of miR221/222 in cancer prognosis. Although the review is quite interesting, I still have some questions. 1) Table 2, it looks like miR-221 expression level only related to liver cancer and NSCLC, which are studied in small sample size (table 1). Is the patient number a decisivefactor in this research? How about the other cancers at a smaller sample size? 2) The pathologic stage and clinic treatment of each patient is unknown, which is closely related to the miR221/222 expression level 3) table 4, miR222 expression level is closely related to 4 cancer types in frozen tissues, but not in FFPE ones, which greatly reduced the studied cases. More study cases should be included to support the conclusion.
Author Response
The manuscript presented by Ravegnini et al provided a review and Meta analysis of miR-221 and miR-222 in cancer prognosis, which could be of interest to the clinic researchers. Although miR221/222 has been recognized as tumor marker, either tumor suppressor or oncogene, this review is quite different from the others. The Meta analysis could provide direct evidence of miR221/222 in cancer prognosis. Although the review is quite interesting, I still have some questions.
1) Table 2, it looks like miR-221 expression level only related to liver cancer and NSCLC, which are studied in small sample size (table 1). Is the patient number a decisive factor in this research? How about the other cancers at a smaller sample size?
Response. We thank the reviewer for the comments.
As reported in table 3 (Previously Table 2, Summary of random-effect meta-analyses for the association between miR-221 overexpression and overall survival in cancer patients) we observed a significant association with poor OS in liver cancer and NSCLC. As highlighted by the reviewer the result is based on a small sample size. The rationale for a meta-analysis is that, by combining the samples of the individual studies, the overall sample size is increased, thereby improving the statistical power of the analysis as well as the precision of the estimates and of effects. Therefore, patients’ number is a critical factor in achieving a robust result, and we cannot exclude that the observed association are chance finding.
2) The pathologic stage and clinic treatment of each patient is unknown, which is closely related to the miR221/222 expression level.
Response. We added cancer stage or grade in Table 1.
3) Table 4, miR222 expression level is closely related to 4 cancer types in frozen tissues, but not in FFPE ones, which greatly reduced the studied cases. More study cases should be included to support the conclusion.
Response. As reported in table 5 (previously table 4, Summary of random-effect meta-analyses for the association between miR-222 overexpression and overall survival in cancer patients) we observed a significant association with poor OS in all cancer subtypes but Urogenital. We are aware that the observed association could be a chance finding; however, to the best of our knowledge, we included in the meta-analysis all the studies available at the time of the analysis. We added a sentence in the conclusion section that emphasizes the possibility that this could be a chance finding, and the need of more powered studies.
Reviewer 2 Report
In this manuscript titled “Prognostic role of miR-221 and miR-222 expression in cancer patients: a systematic review and meta-analysis”, Ravegnini et al. performed a meta-analysis to investigate the prognostic value of miR-221, 222 or the cluster miR-221/222 in different cancers. Previously, other research groups performed similar investigation; showing controversial results. Here, Ravegnini et al. conducted an updated and wider analysis to provide new findings and to fill the gap on this topic.
By using a combination of key words, a systematic review was conducted on three databases; this first screening identified 852 potential studies among these, 50 were finally eligible for the meta-analysis. The current study claimed that the high expression of both miR-221 and miR-222 is a predictive factor of poor OS in cancer; on the contrary, the results obtained from subgroups analysis were less robust, in particular for miR-221.
The methods section was well described and the analysis rigorously performed.
Comments:
Introduction
In general, the introduction needs more proof reading to improve the tone of the manuscript. In detail, some sentences are a bit confusing and can be rewritten (Line 46 to 48. The authors stated: "…giving their central role in basically all the cellular processes, it is clear that deregulation of miRNAs may promote disruption of a tiny equilibrium within a specific cell, altering physiologically relevant functions". This sentence should be clarified and perhaps include more references. It is unclear what "specific cell" means. Line 52: "the most recently described miRNA sponges". This sentence seems a bit awkward. Line 68-line 69: "In addition every single study may be underpowered to achieve a comprehensive and reliable conclusion". It is not clear the meaning of this sentence).
Methods
Subgroups: authors analysed several subgroups, such as patient ethnicity, tumor type...etc.
More subgroups should be considered (where possible): gender, stage of cancer, received or non-received treatment.
Results
Figure 1: a top section should be added where all 852 studies are included and describing the criteria of selection (as authors did it in the main text, section materials and methods -inclusion and exclusion criteria)
Study characteristics and quality assessment’ section: line 151: more detail regarding each study should be provided (median age of patients, gender and stage of cancer) and included in table 1.
Discussion
Line 300 to 303: The authors highlighted some limitations of the study linked to patients’ characteristics, treatment received and duration of follow-up. Here, references should be added, emphasizing how the expression of miR-221 or miR-222 may change under those different circumstances.
In general, the authors should highlight more the novelty of this manuscript and the new findings (if any) compared to other similar meta-analysis (see: Zhang, P.; Zhang, M.; Han, R.; Zhang, K.; Ding, H.; Liang, C.; Zhang, L. The Correlation between MicroRNA-221/222 Cluster Overexpression and Malignancy: An Updated Meta-Analysis Including 2693 Patients. Cancer Manag. Res. 2018, Volume 10, 3371–3381. https://doi.org/10.2147/CMAR.S171303. Wang, J.; Liu, S.; Sun, G. P.; Wang, F.; Zou, Y. F.; Jiao, Y.; Ning, J.; Xu, J. Prognostic Significance of MicroRNA-221/222 Expression in Cancers: Evidence from 1,204 Subjects. Int. J. Biol. Markers 2014, 29 (2), 129–141. https://doi.org/10.5301/jbm.5000058).
Conclusion
Line 319: “Furthermore, the elucidation of the molecular mechanism of miR-222 in cancer prognosis will lead to improvements in cancer patients’ management.” This sentence sounds out of context.
Author Response
In this manuscript titled “Prognostic role of miR-221 and miR-222 expression in cancer patients: a systematic review and meta-analysis”, Ravegnini et al. performed a meta-analysis to investigate the prognostic value of miR-221, 222 or the cluster miR-221/222 in different cancers. Previously, other research groups performed similar investigation; showing controversial results. Here, Ravegnini et al. conducted an updated and wider analysis to provide new findings and to fill the gap on this topic.
By using a combination of key words, a systematic review was conducted on three databases; this first screening identified 852 potential studies among these, 50 were finally eligible for the meta-analysis. The current study claimed that the high expression of both miR-221 and miR-222 is a predictive factor of poor OS in cancer; on the contrary, the results obtained from subgroups analysis were less robust, in particular for miR-221.
The methods section was well described and the analysis rigorously performed.
Response: we thank the reviewer for the comments.
Comments:
Introduction
1. In general, the introduction needs more proof reading to improve the tone of the manuscript. In detail, some sentences are a bit confusing and can be rewritten:
- Line 46 to 48. The authors stated: "…giving their central role in basically all the cellular processes, it is clear that deregulation of miRNAs may promote disruption of a tiny equilibrium within a specific cell, altering physiologically relevant functions". This sentence should be clarified and perhaps include more references. It is unclear what "specific cell" means.
Response. We reformulated the sentence, adding more references.
1C. Line 52: "the most recently described miRNA sponges". This sentence seems a bit awkward.
Response. We reformulated the sentence, clarifying the meaning.
1D. Line 68-line 69: "In addition every single study may be underpowered to achieve a comprehensive and reliable conclusion". It is not clear the meaning of this sentence.
Response. We reformulated the sentence, clarifying the meaning.
Methods
1. Subgroups: authors analyzed several subgroups, such as patient ethnicity, tumor type...etc. More subgroups should be considered (where possible): gender, stage of cancer, received or non-received treatment.
Response. With regard to subgroups analysis, we included ethnicity and tumor type, which were obtainable from the paper. With regard to stage/grade, we added this information within table 1. However, we did not include this information in subgroups analysis as survival data where not extractable according to stage. With regard to gender, the information is available as a demographic parameter (added in the new table 1), we do not have survival analysis according to gender therefore we could not perform a subgroup analysis. With regard to treatment, we could add information about pre-surgery treatment. We included the information in table 1 and in the subgroups analysis. The text has been modified accordingly.
Results
Figure 1: a top section should be added where all 852 studies are included and describing the criteria of selection (as authors did it in the main text, section materials and methods -inclusion and exclusion criteria).
Response. We added a top section in figure 1.
Study characteristics and quality assessment’ section: line 151: more detail regarding each study should be provided (median age of patients, gender and stage of cancer) and included in table 1.
Response. The requested information have been added in table 1.
Discussion
Line 300 to 303: The authors highlighted some limitations of the study linked to patients’ characteristics, treatment received and duration of follow-up. Here, references should be added, emphasizing how the expression of miR-221 or miR-222 may change under those different circumstances.
Response. This limitations have been highlighted in many papers . We added references
In general, the authors should highlight more the novelty of this manuscript and the new findings (if any) compared to other similar meta-analysis (see: Zhang, P.; Zhang, M.; Han, R.; Zhang, K.; Ding, H.; Liang, C.; Zhang, L. The Correlation between MicroRNA-221/222 Cluster Overexpression and Malignancy: An Updated Meta-Analysis Including 2693 Patients. Cancer Manag. Res. 2018, Volume 10, 3371–3381. https://doi.org/10.2147/CMAR.S171303. Wang, J.; Liu, S.; Sun, G. P.; Wang, F.; Zou, Y. F.; Jiao, Y.; Ning, J.; Xu, J. Prognostic Significance of MicroRNA-221/222 Expression in Cancers: Evidence from 1,204 Subjects. Int. J. Biol. Markers 2014, 29 (2), 129–141. https://doi.org/10.5301/jbm.5000058).
Response. the previous meta-analysis included a lower number of patients and present limitations; therefore, our study, as stated in the text, represents an important up-date on the topic. We also discussed limitations of this first two meta-analysis within the discussion.
Conclusion
Line 319: “Furthermore, the elucidation of the molecular mechanism of miR-222 in cancer prognosis will lead to improvements in cancer patients’ management.” This sentence sounds out of context.
Response. We agree; we deleted this sentence.
Reviewer 3 Report
The current study authored by Ravegnini et.al. has investigated the prognostic role of miR-221 and miR-222 expression in cancer patients. For these purpose authors performed a systematic review and meta-analysis of previously published data and investigated the effects of miR-221/222 on overall survival (OS) and other secondary outcomes among cancer patients. Authors found that high expression of miR-222 is associated with a poor prognosis in cancer patients, whereas significance of miR-221 remains unclear to OS to the cancer patients. However, this systemic reviewer and meta-analysis has several major concerns that authors should address further.
1. People have already shown the association of miR-221/222 with different cancer. So, authors should discus more specifically why this study is important or what were the drawback of the previous studies and how the current analysis/study overcome all the drawback from the previous studies.
a. Curr Mol Med. 2012 Jan; 12(1): 27–33.
b. DOI: 10.7754/Clin.Lab.2016.160102
c. doi: 10.3389/fimmu.2017.00056
2. All the data analyzed here based on the cellular miRNA. It would be better and more convincing to use the circulating miRNA data also. Authors should include the circulating miRNA data in their analysis. Such as:
a. http://www.cancerjournal.net/text.asp?2019/15/1/115/244469
b. DOI: 10.1007/s12032-014-0164-8
3. It would be better to discuss the functional role of the miRNAs with the cancers also or other aspects like treatment.
4. Most importantly, the study should discuss about the different stages of the cancers and their relation with miRNA expression and the analysis the OS with the different stages.
5. English has to be polished.
Author Response
The current study authored by Ravegnini et.al. has investigated the prognostic role of miR-221 and miR-222 expression in cancer patients. For these purpose authors performed a systematic review and meta-analysis of previously published data and investigated the effects of miR-221/222 on overall survival (OS) and other secondary outcomes among cancer patients. Authors found that high expression of miR-222 is associated with a poor prognosis in cancer patients, whereas significance of miR-221 remains unclear to OS to the cancer patients. However, this systemic reviewer and meta-analysis has several major concerns that authors should address further.
1. People have already shown the association of miR-221/222 with different cancer. So, authors should discus more specifically why this study is important or what were the drawback of the previous studies and how the current analysis/study overcome all the drawback from the previous studies. a. Curr Mol Med. 2012 Jan; 12(1): 27–33; b. DOI: 10.7754/Clin.Lab.2016.160102; c. doi: 10.3389/fimmu.2017.00056.
Response. We thank the reviewer for the comments. As stated in the text, our meta-analysis, including more studies than the previous two, represents an important up-date on the topic. With regard to the work in Curr Mol Med and the work with doi 10.3389/fimmu.2017.00056, these are both review and not meta-analysis; therefore the meaning of these works are different.
2. All the data analyzed here based on the cellular miRNA. It would be better and more convincing to use the circulating miRNA data also. Authors should include the circulating miRNA data in their analysis. Such as: a. http://www.cancerjournal.net/text.asp?2019/15/1/115/244469; b. DOI: 10.1007/s12032-014-0164-8
Response. We did not add studies on circulating miRNA, as we think that this analysis should be done separately from the cellular (tissue) miRNAs. Indeed, the first step is to verify if the two measures are comparable, and therefore if they can be combined. The number of papers on circulating tumor is rapidly increasing, we do expect soon a meta-anlaysis considering the circulating miRNAs and a comparative analysis with the results obtained from tissue.
3. It would be better to discuss the functional role of the miRNAs with the cancers also or other aspects like treatment. Response.The functional role of mir221/222 in cancer and tumor treatment have been extensively reported in several papers. We added those references.
4. Most importantly, the study should discuss about the different stages of the cancers and their relation with miRNA expression and the analysis the OS with the different stages.
Response. We added stages in table 1. However, we did not include this information in subgroups analysis as survival data were not extractable according to stage. In any cases, as stated in the conclusion, we recognize that the impossibility to correlates survival parameters with tumor stages could represent a limit of the study.
5. English has to be polished.
Response. We checked English language using freely available on-line tools.